# Lipid Metabolism Alterations in Hyperlipidemic Dogs with Biliary Tract or Endocrine Diseases

**DOI:** 10.3390/ani15020256

**Published:** 2025-01-17

**Authors:** Tae-Woo Kim, Min-Hee Kang, Hee-Myung Park

**Affiliations:** 1Department of Veterinary Internal Medicine, College of Veterinary Medicine, Konkuk University, Seoul 05029, Republic of Korea; epn5261@naver.com; 2Laboratory of Veterinary Pathology, College of Veterinary Medicine, Chungbuk National University, Cheongju 28644, Republic of Korea; 3Department of Bio-Animal Health, Jangan University, Hwaseong 18331, Republic of Korea; mhkang@jangan.ac.kr

**Keywords:** biliary tract disease, endocrine disease, hyperlipidemia, lipid metabolism, lipoprotein electrophoresis

## Abstract

This retrospective study investigated lipid metabolism alterations in 65 dogs with hyperlipidemia using lipoprotein electrophoresis (LPE). The analysis revealed that dogs with biliary tract diseases or underlying metabolic disorders, such as endocrine diseases, exhibited higher pre-beta fractions and cholesterol concentrations compared to non-hyperlipidemic dogs. Significant linear relationships were identified between serum gamma-glutamyl transpeptidase (GGT) levels and both pre-beta fractions and cholesterol concentrations, highlighting the impact of biliary tract dysfunction on lipid metabolism. Additionally, Miniature Schnauzers, a breed predisposed to hyperlipidemia, showed distinct lipid profile abnormalities, including reduced alpha fractions and elevated triglyceride and cholesterol levels. These findings highlight the clinical relevance of LPE as a useful tool for identifying lipid metabolism abnormalities and monitoring disease-related changes. Veterinary practitioners should consider altered lipid metabolism, including specific ratios such as the GGT/ALT and GGT/ALP ratios, as potential indicators of biliary tract or endocrine diseases when evaluating hyperlipidemic patients in small animal practice. These findings emphasize the importance of tailored diagnostic and therapeutic strategies to address the underlying metabolic disorders.

## 1. Introduction

Lipids, including triglycerides and cholesterol, are integral to numerous physiological processes in living organisms, serving as key components of cell membranes, energy reserves, and signaling molecules [1,2]. Due to their hydrophobic nature, lipids are transported in plasma through macromolecular complexes known as lipoproteins [2,3]. These lipoproteins are categorized into four major classes based on their size, density, and chemical composition: chylomicrons, very low-density lipoproteins (VLDLs), low-density lipoproteins (LDLs), and high-density lipoproteins (HDLs) [2,4]. Each class exhibits distinct electrophoretic mobility, with chylomicrons and VLDLs migrating to the pre-beta position, LDLs to the beta position, and HDLs to the alpha position on lipoprotein electrophoresis (LPE) gels [4,5,6]. In healthy dogs, HDL predominates as the primary lipoprotein fraction identified by LPE [6].

The liver plays a central role in lipid metabolism, synthesizing triglycerides and cholesterol from acetyl-CoA [7]. Excess cholesterol from dietary and hepatic sources is either excreted via bile or secreted into circulation as VLDL particles. These VLDL particles are hydrolyzed by lipoprotein lipase (LPL), which releases triglycerides to peripheral tissues, forming cholesterol-enriched LDL [2,7]. LDL receptors facilitate the uptake of LDL, enabling cholesterol delivery to cells and eventual clearance by the liver [7].

Hyperlipidemia refers to elevated levels of lipids, particularly triglycerides (hypertriglyceridemia) and/or cholesterol (hypercholesterolemia), in fasting blood samples [4,8]. This condition may result from genetic predispositions, such as in Miniature Schnauzers, or from metabolic disorders [8,9]. Extrahepatic cholestasis, characterized by bile duct obstruction, can impair bile flow, causing regurgitation of cholesterol-laden bile into the liver, which often results in hypercholesterolemia and hypertriglyceridemia [2,10]. Similarly, endocrine disorders can disrupt lipid clearance and lipoprotein metabolism, leading to elevated VLDL production and altered LDL/HDL uptake by tissues [1,7,8].

The purpose of this retrospective study was to analyze the characteristic variations in lipoprotein electrophoresis (LPE) in hyperlipidemic dogs and investigate the associations between lipid metabolism alterations and underlying biliary tract or endocrine diseases.

## 2. Materials and Methods

### 2.1. Case Selection

Clinical records of 65 dogs that underwent lipoprotein electrophoresis (LPE) at the Veterinary Medical Teaching Hospital (VMTH) of Konkuk University between January 2007 and August 2019 were reviewed. Data collected included signalment, medical history, physical examination findings, biochemical profiles, laboratory test results, and LPE findings.

### 2.2. Inclusion and Exclusion Criteria

Sixty-five dogs that underwent LPE as part of routine diagnostic testing to gather initial clinical data for differential and definitive diagnoses were included in the study. LPE testing was performed on dogs with a body condition score (BCS) greater than 4/5 (five-point system) or when blood screening tests indicated hyperlipidemia, such as hypertriglyceridemia and/or hypercholesterolemia. The recorded LPE patterns were used to support diagnostic assessments.

Dogs were excluded if they were pregnant, lactating, intended for breeding, or less than one year of age, as these conditions are known to influence lipid metabolism [11]. Additionally, dogs that underwent biochemical testing within 7 to 12 h after a meal were excluded, as non-fasting status can result in postprandial hyperlipidemia, a normal physiological state [8].

### 2.3. Signalment and Clinical Presentations

Signalment, including breed, age, sex, BCS, and neutering status, was documented. Clinical presentations, including anorexia, vomiting, diarrhea, abdominal distention, abdominal pain, neurologic signs (e.g., seizures, peripheral neuropathy), ophthalmologic signs (e.g., lipemia retinalis), and cutaneous signs (e.g., xanthomas, lipomas), were evaluated [2,7].

### 2.4. Biochemical Profile Evaluation

A complete blood count (CBC) was performed using the VetScan^®^ HM2 Hematology System (Abaxis, Inc., Union City, CA, USA), and serum chemistry analysis was conducted with the Cobas C III instrument (Roche Diagnostics, South San Francisco, CA, USA). Electrolyte analysis was carried out using the 9180 Electrolyte Analyzer (Roche Diagnostics, South San Francisco, CA, USA). These tests were used to evaluate the general condition of the patients, including renal and hepatic function, and to identify any underlying diseases.

Additional diagnostic tests, including thyroid hormone profiling, canine pancreatic lipase (cPLI), canine trypsin-like immunoreactivity (cTLI), urine protein-to-creatinine ratio (UPC), ACTH stimulation testing, and D-dimer measurement, were performed at the Neodin Veterinary Science Institute (Seobinggo-dong, Yongsan-gu, Seoul, Republic of Korea) and ANTECH Diagnostics (Irvine, CA, USA).

### 2.5. Evaluation of Underlying Diseases

The final diagnosis was determined based on findings from physical examinations, hematologic tests, biochemistry panels, urinalysis, radiographic imaging, and ultrasonography. Hyperlipidemia was identified by elevated serum triglyceride (TG) levels (reference range: 19–133 mg/dL), cholesterol levels (reference range: 135–345 mg/dL), or both. Dogs were classified as hyperlipidemic if TG levels were elevated (hypertriglyceridemia), cholesterol levels were elevated (hypercholesterolemia), or elevations were observed in both parameters.

Biliary tract diseases were diagnosed when serum gamma-glutamyl transferase (GGT) levels (reference range: 0–6 IU/L) and alkaline phosphatase (ALP) levels (reference range: 15–127 IU/L) were elevated, supported by ultrasonographic findings consistent with biliary tract abnormalities, such as bile duct dilatation, thickening of the bile duct wall and mucocele, and elevated total bilirubin levels (T. Bil, reference range: 0~0.344 (mg/dL)) [12], indicating extrahepatic hyperbilirubinemia. In addition, the GGT/ALT ratio, GGT/ALP ratio, and AST/ALT ratio, clinically useful liver function tests (LFTs) in humans for differentiating liver problems from bile duct issues, were calculated and evaluated in this study to distinguish between biliary tract diseases and liver-specific conditions when hepatobiliary parameters were elevated [13,14,15].

In diagnosing endocrine disorders, Hyperadrenocorticism is identified through an ACTH Stimulation Test, with post-ACTH cortisol levels exceeding the reference range (5.50–20.00 µg/dL). Hypothyroidism is diagnosed based on changes in the thyroid profile, such as low total T4 (reference range: 0.8–3.5 µg/dL), low free T4 (reference range: 8–40 pmol/L), and elevated TSH (reference range: 0.00–0.60 ng/mL). Diabetes mellitus is evaluated through serum chemistry and urinalysis alongside history, which commonly includes polyuria, polydipsia, obesity, hyperglycemia, hyperlipidemia, and glucosuria.

In addition, pancreatitis was diagnosed based on elevated cPLI levels (reference range: 0–200 µg/dL) and/or cTLI levels (reference range: 5–35 µg/dL) in the quantitative analysis. Protein-losing nephropathy (PLN) was diagnosed when UPC levels exceeded 0.5 [16].

For cases involving multiple diagnoses, the final diagnosis was categorized based on the primary clinical symptoms and overall condition of the patient.

### 2.6. Methodology and Evaluation of Lipoprotein Electrophoresis (LPE)

Blood samples were collected by venipuncture into serum separation tubes (SSTs) from dogs that had fasted for over 12 h. The samples were stored at −20 °C and transported to the Neodin Veterinary Science Institute. Serum analysis was conducted using the HYDRASYS^®^ agarose gel electrophoresis system (Sebia, Norcross, GA, USA), following the manufacturer’s protocol. The samples were stained with 0.1% Fat Red, and gel images were scanned with a GELSCAN densitometer (Sebia, Norcross, GA, USA). Data analysis was performed using PHORESIS^TM^ software (version 2.0).

The dogs were categorized into two primary groups based on their hyperlipidemic status: the non-hyperlipidemic (NHL) group and the hyperlipidemic (HL) group. The HL group was further divided into three subgroups:1.H subgroup: hyperlipidemic dogs without biliary tract or endocrine diseases;2.H + B subgroup: hyperlipidemic dogs with biliary tract diseases but without endocrine diseases;3.H + B + E subgroup: hyperlipidemic dogs with both biliary tract and endocrine diseases.

Endocrine diseases considered in this study included hypothyroidism, hyperadrenocorticism, and diabetes mellitus, which are known to affect lipid metabolism. Additionally, patients were classified according to underlying metabolic disturbances, such as obesity, hypothyroidism, hyperadrenocorticism, protein-losing nephropathy, diabetes mellitus, and pancreatitis [1,2,7,11]. Miniature Schnauzers, a breed predisposed to primary hyperlipidemia, were analyzed separately as a single group of 18 dogs.

LPE results provided detailed information on lipoprotein fractions, including the proportions of alpha, beta, and pre-beta fractions, and the concentrations of chylomicrons, triglycerides, and cholesterol. In the absence of established reference ranges for LPE fractions, the results were evaluated through intergroup statistical comparisons of lipoprotein fractions and analyses between the NHL group and groups with specific metabolic disturbances.

### 2.7. Statistical Analysis

All the data were statistically analyzed by cross-tabulation, including the Chi-square (χ^2^) test for qualitative variables, independent *t*-test, and one-way analysis of variance (ANOVA). Post hoc comparisons were conducted using Scheffé’s test to evaluate differences between groups based on the normality of data distribution. Correlation and regression analyses were performed to examine the linear relationship between each lipoprotein fraction and serum biochemistry values. All statistical analyses were conducted using SPSS Statistics 20.0 software, with a significance threshold set at *p* < 0.05.

## 3. Results

### 3.1. Signalment

The signalment of the 65 dogs included in this study is summarized in Table 1. Dogs aged 5 to 10 years comprised the majority of the HL group (*n* = 32; 63.5%), followed by those older than 10 years (*n* = 13; 26.5%) and younger than 5 years (*n* = 4; 8.2%). A statistically significant difference was observed among the groups regarding breed distribution (*p* = 0.026).

In the case of BCS score, there were relatively more animals with a BCS greater than 4/5 (five-point system) in the NHL group (*n* = 13; 81.2%) than in the HL group (*n* = 35; 71.5%), but no significance was observed in the chi-square test.

Within the HL group, the most commonly represented breed was the Miniature Schnauzer (*n* = 18; 27.7%), followed by the Yorkshire Terrier (*n* = 6; 12.2%), Shih Tzu (*n* = 6; 12.2%), Maltese (*n* = 4; 8.2%), mixed-breed dogs (*n* = 4; 8.2%), Chihuahua (*n* = 2; 4.1%), and other breeds (*n* = 7; 10.8%).

### 3.2. Clinical Presentations

Table 2 and Table 3 summarize the clinical presentations of dogs in the NHL and HL groups. In terms of clinical signs, abdominal distension was significantly more frequent in the HL group (26.5%) compared to the NHL group (0%; *p* = 0.021). While vomiting, diarrhea, abdominal pain, and cutaneous signs (e.g., xanthomas and lipomas) were observed more frequently in the HL group than in the NHL group, these differences were not statistically significant.

In clinical diagnosis, the frequency of hyperadrenocorticism (HAC), protein-losing nephropathy (PLN), diabetes mellitus (DM), and pancreatitis was higher in the HL group than that of NHL, while the frequency of obesity and hyperthyroidism was higher in the NHL group versus the HL group. However, these differences were not statistically significant.

### 3.3. Hematologic and Serum Chemistry Evaluation

The hematologic profiles of the NHL and HL groups are summarized in Table 4. Parameters including white blood cell count (WBC), red blood cell count (RBC), hemoglobin (HGB), mean corpuscular volume (MCV), mean corpuscular hemoglobin concentration (MCHC), and platelet count (PLT) showed no statistically significant differences between the two groups.

The serum chemistry and electrolyte profiles of the NHL and HL groups are presented in Table 5. Total cholesterol levels were significantly higher in the HL group (389.67 ± 172.57 mg/dL) compared to the NHL group (231.69 ± 62.37 mg/dL; *p* = 0.001). Triglyceride levels were also markedly elevated in the HL group (435.78 ± 449.64 mg/dL) compared to the NHL group (89.56 ± 31.60 mg/dL; *p* = 0.003).

While ALT, AST, ALP, GGT, LDH, total bilirubin (T. Bil), glucose, and lipase levels were higher in the HL group than in the NHL group, these differences were not statistically significant. Similarly, total protein (TP), albumin (ALB), ammonia (NH3), sodium (Na), potassium (K), and chloride (Cl) levels did not differ significantly between the two groups.

### 3.4. Evaluation of Hepatobilliary Profiles

The hepatobiliary profiles assessed for underlying biliary disease for the NHL group and the three subgroups of the HL group are presented in Table 6 and Figure 1. ALT, ALP, and GGT levels showed a statistically significant tendency to be higher in the H + B subgroup and H + B + E subgroup compared to the NHL group and H subgroup. Additionally, the GGT/ALT ratio (GGT/ALT) demonstrated a statistically significant increase in the NHL group and H + B subgroup compared to the H subgroup, while the AST/ALT ratio (AST/ALT) showed a statistically significant increase in the H + B subgroup compared to the NHL group and H subgroup.

### 3.5. Comparison of Lipid Profiles Between Groups

The lipid fraction profiles assessed by LPE for the NHL group and the three subgroups of the HL group are presented in Figure 2 and Table 7. Each lipoprotein fraction displayed a distinct trend, with the alpha fraction decreasing and the pre-beta fraction increasing progressively from the NHL group to the H + B + E subgroup of the HL group.

The alpha fraction in the H + B subgroup of the HL group (48.16 ± 14.16%) was significantly lower than that of the NHL group (62.50 ± 14.17%; *p* = 0.048). Similarly, the pre-beta fraction in the H + B + E subgroup of the HL group (48.16 ± 16.51%) was significantly higher compared to the H + B subgroup (25.21 ± 13.27%), the H subgroup (22.63 ± 15.87%), and the NHL group (19.13 ± 8.56%; *p* = 0.040).

For the beta fraction, levels in the H + B subgroup (26.84 ± 12.67%) and the H subgroup (21.17 ± 15.32%) were higher compared to the H + B + E subgroup (14.50 ± 6.44%) and the NHL group (18.50 ± 11.94%), but these differences were not statistically significant.

Triglyceride concentrations were significantly elevated in the H + B subgroup (458.53 ± 369.40 mg/dL) and the H subgroup (449.71 ± 551.76 mg/dL) compared to the NHL group (89.56 ± 31.60 mg/dL; *p* = 0.026). Cholesterol concentrations were also significantly higher in the H subgroup (364.92 ± 210.88 mg/dL), H + B subgroup (408.32 ± 133.40 mg/dL), and H + B + E subgroup (429.67 ± 105.34 mg/dL) than in the NHL group (231.69 ± 62.37 mg/dL; *p* = 0.006).

### 3.6. Linear Relationships Between Lipid Profiles and Biochemical Values Related to Lipid and Hepatobiliary Disease

Linear relationships between lipid profiles, including lipoprotein fractions, triglycerides, and cholesterol levels, and serum biochemical markers associated with hepatobiliary diseases (ALT, AST, ALP, GGT, LDH, and T. Bil) were evaluated using correlation and regression analyses.

The analysis revealed a significant positive correlation between GGT levels and the pre-beta fraction, indicating that higher GGT levels were correlated with an increase in the pre-beta fraction (regression equation: pre-beta fraction = 22.115 + GGT × 0.108; correlation coefficient: 0.257; *p* = 0.039). Similarly, GGT levels showed a significant positive correlation with cholesterol levels, where elevated GGT levels corresponded to higher cholesterol concentrations (regression equation: cholesterol = 325.105 + GGT × 1.506; correlation coefficient: 0.307; *p* = 0.039). The GGT/ALT ratio showed a significant positive correlation with cholesterol levels, where the elevated GGT/ALT ratio corresponded to higher cholesterol concentrations (regression equation: cholesterol = 324.629 + GGT/ALT ratio × 101.871; correlation coefficient: 0.298; *p* = 0.016). The AST/ALT ratio showed a significant positive correlation with cholesterol levels, where the elevated GGT/ALT ratio corresponded to higher cholesterol concentrations (regression equation: triglyceride = 274.963 + AST/ALT ratio × 100.832; correlation coefficient: 0.275; *p* = 0.027).

Additionally, LDH levels demonstrated a significant linear relationship with triglyceride concentrations, suggesting that increases in LDH levels were associated with elevated triglyceride levels (regression equation: triglyceride = 273.693 + LDH × 0.631; correlation coefficient: 0.268; *p* = 0.031). These findings are illustrated in Figure 3.

### 3.7. Lipid Profiles and Underlying Diseases

The lipid fraction profiles obtained from serum lipoprotein electrophoresis in the NHL group and dogs with underlying conditions, including obesity, hypothyroidism, hyperadrenocorticism (HAC), protein-losing nephropathy (PLN), diabetes mellitus (DM), and pancreatitis, are presented in Table 8.

The alpha fraction of lipoproteins was lower in groups with underlying diseases compared to the NHL group; however, this difference was not statistically significant. The pre-beta fraction was significantly higher in dogs with hypothyroidism (33.25 ± 16.01%) and hyperadrenocorticism (39.50 ± 6.61%) compared to the NHL group (19.10 ± 8.56%) and those with obesity (18.44 ± 11.87%) or pancreatitis (19.25 ± 9.81%; *p* = 0.008).

The beta fraction and triglyceride concentrations were elevated in all groups with underlying diseases, except the hypothyroidism group, relative to the NHL group; however, these differences were not statistically significant. Cholesterol concentrations were significantly higher in dogs with obesity (307.16 ± 135.63 mg/dL), hyperadrenocorticism (414.25 ± 137.97 mg/dL), diabetes mellitus (415.20 ± 120.76 mg/dL), and pancreatitis (375.25 ± 133.86 mg/dL) compared to the NHL group (231.69 ± 62.37 mg/dL; *p* = 0.032).

### 3.8. Lipid Profiles of Schnauzers

Lipid profiles, including lipoprotein fractions, triglyceride levels, and cholesterol concentrations, were analyzed in 18 Miniature Schnauzers, a breed known to be predisposed to primary hyperlipidemia (Table 9). All Schnauzers in this study were classified within the HL group due to their hyperlipidemic status. Elevated beta fraction levels were observed in 3 of the 18 Schnauzers, while elevated pre-beta fraction levels were noted in another three Schnauzers.

Ten of the 18 Schnauzers were assigned to the H subgroup of the HL group, as they exhibited hyperlipidemia without concurrent biliary tract or endocrine diseases. Among these 10 Schnauzers, the alpha fraction was below 50%, while the combined pre-beta and beta lipoprotein fractions exceeded 50%.

The mean alpha fraction in the 18 Schnauzers (49.22 ± 15.22%) was significantly lower than that in the NHL group (62.50 ± 3.54%; *p* = 0.013). The pre-beta and beta fractions in the Schnauzers (25.00 ± 14.37% and 25.78 ± 16.05%, respectively) were higher than those in the NHL group (19.13 ± 2.14% and 18.50 ± 2.99%, respectively), although these differences were not statistically significant (*p* = 0.164 and *p* = 0.148, respectively).

Triglyceride (606.83 ± 569.12 mg/dL) and cholesterol concentrations (369.94 ± 115.85 mg/dL) were significantly higher in the 18 Schnauzers compared to the NHL group (89.56 ± 7.89 mg/dL and 231.69 ± 15.59 mg/dL, respectively; *p* = 0.001 and *p* < 0.001, respectively).

## 4. Discussion

This study investigated the alterations in lipid profiles associated with hyperlipidemia in dogs, focusing on biliary tract diseases, endocrine disorders, and the predisposed Miniature Schnauzer breed. Using lipoprotein electrophoresis (LPE), we analyzed lipoprotein fractions and lipid concentrations, providing valuable insights into the intricate relationship between lipid metabolism and various disease conditions. The findings revealed significant differences in lipoprotein patterns and lipid concentrations, highlighting the potential clinical utility of LPE as a diagnostic and monitoring tool in small animal practice.

Hyperlipidemia, particularly fasting hyperlipidemia, is a valuable diagnostic marker in veterinary medicine, indicative of potential underlying conditions and influencing serum biochemistry results. Consistent with previous studies, our results support the association between hyperlipidemia and secondary illnesses such as acute pancreatitis [1,7,16]. In the HL group, significant elevations in pre-beta fractions and cholesterol levels were observed compared to the NHL group. Furthermore, statistically significant correlations between GGT and GGT/ALT, a biliary biomarker, and lipid fractions strengthen the hypothesis that biliary tract diseases contribute to hypercholesterolemia through mechanisms such as impaired cholesterol clearance and altered lipoprotein metabolism [2,12,13,14,15,17,18,19].

Biliary tract diseases, including cholestasis, are well-documented contributors to abnormal lipid profiles. Previous studies have demonstrated that regurgitation of biliary lipids into the liver can lead to excessive cholesterol pools, downregulating LDL receptor activity and disrupting lipoprotein balance [1,9]. These mechanisms result in elevated cholesterol levels and altered lipoprotein fractions, as seen in our study. Dogs with biliary tract diseases in the HL group exhibited increased pre-beta fractions and cholesterol concentrations, suggesting that hyperlipidemia may serve as an indicator of biliary tract dysfunction and related complications.

In human medicine, liver function tests (LFTs) such as the GGT/ALT ratio, GGT/ALP ratio, and AST/ALT ratio are used to determine whether specific increases in liver parameters are associated with liver issues, bile duct problems, or specific diseases like hepatitis virus infection, liver tumors, or alcohol-dependent liver disease [13,14,15]. In this study, while the GGT/ALT and AST/ALT ratios were significantly elevated in the H + B subgroup with biliary disease compared to the H subgroup without biliary disease, there was no significant change in the H + B + E subgroup with both biliary and endocrine diseases, which was a limitation. However, distinguishing whether increases in liver-related values are due to liver issues or bile duct problems remains critical for diagnosis in the veterinary field as well. Therefore, as these discoveries accumulate and the ratios become more widely available in the veterinary field, these ratios are expected to provide valuable evidence for veterinarians when dealing with underlying diseases such as metabolic changes and biliary diseases.

Endocrine disorders such as hypothyroidism, hyperadrenocorticism, and diabetes mellitus are known to affect lipid metabolism through decreased activity of lipoprotein lipase (LPL). LPL is critical for the hydrolysis of chylomicrons and VLDL, facilitating triglyceride clearance and fatty acid delivery to tissues. Impaired LPL activity results in hypertriglyceridemia and elevated pre-beta fractions, as observed in this study [7]. Groups with hypothyroidism, hyperadrenocorticism, diabetes mellitus, and pancreatitis exhibited elevated cholesterol concentrations and pre-beta fractions compared to the NHL group. These findings align with prior research indicating that endocrine disorders exacerbate lipid metabolism abnormalities through disrupted LPL and LDL receptor activity [1,2,11,20].

Eighteen Miniature schnauzers, a breed known to have hyperlipoproteinemia caused by deficient LPL activity, were analyzed for lipid content and lipoprotein fractions [20,21]. All schnauzers were classified as the HL group based on their hyperlipidemic status, and the predominant lipoprotein fractions of 10 of the 18 Schnauzers were elevated pre-beta and/or beta fractions, as opposed to the alpha fractions typically observed in the NHL group. In addition, triglycerides and cholesterol levels were significantly higher than in the NHL group. These differences in the lipid profiles of the schnauzer group versus NHL group were in agreement with the changes of previous studies, supporting a tendency of idiopathic familial hyperlipidemia in the Schnauzer [20,22,23]. While the exact reasons for these differences in lipid and lipoprotein profiles remain unclear, it is possible that Schnauzers have a distinct lipoprotein metabolism including LPL activity, compared to other breeds, with only some individuals developing severe lipid metabolism disorders leading to hyperlipidemia [22,23,24].

The findings of this study underscore the clinical relevance of LPE as a useful tool for evaluating lipid metabolism abnormalities. LPE provides semi-quantitative data on lipoprotein fractions, enabling the identification of deviations from normal profiles. Despite the absence of standardized reference ranges, LPE offers valuable insights into the relative changes in lipoprotein composition, particularly reductions in alpha fractions accompanied by increases in pre-beta and beta fractions [3,5,8]. These results can guide clinicians in diagnosing hyperlipidemia caused by genetic factors or secondary metabolic disturbances.

However, this study has limitations. The lack of established reference ranges for lipoprotein fractions restricts direct comparisons with previous studies. Additionally, the relatively small sample size, especially in the groups with endocrine diseases, limits the statistical power of the analysis. Future research involving larger populations and diverse disease conditions is necessary to establish comprehensive reference ranges and to further elucidate the role of lipid metabolism in various disease states. Moreover, evaluating the impact of therapeutic interventions, such as dietary modifications and lipid-lowering treatments, on lipid profiles and clinical outcomes would provide valuable insights for managing hyperlipidemic patients.

## 5. Conclusions

This study demonstrates significant associations between lipid metabolism alterations and biliary tract diseases, endocrine disorders, and breed predisposition in dogs.

The analysis revealed that dogs with biliary tract diseases or metabolic disorders showed higher pre-beta fractions and cholesterol levels. Significant correlations between gamma-glutamyl transpeptidase (GGT) and lipid markers highlighted the role of biliary dysfunction. Miniature Schnauzers, predisposed to hyperlipidemia, displayed reduced alpha fractions and elevated triglyceride and cholesterol levels. Hyperlipidemia, as identified through LPE, can serve as an indicator of underlying metabolic disturbances. Clinicians should consider hyperlipidemia as a diagnostic marker for conditions such as biliary tract diseases and endocrine disorders and should address it through dietary modifications and therapeutic interventions aimed at reducing lipid levels and improving metabolic health. Further research is needed to refine diagnostic criteria and expand the understanding of lipid metabolism in small animal medicine.

## Figures and Tables

**Figure 1 animals-15-00256-f001:**
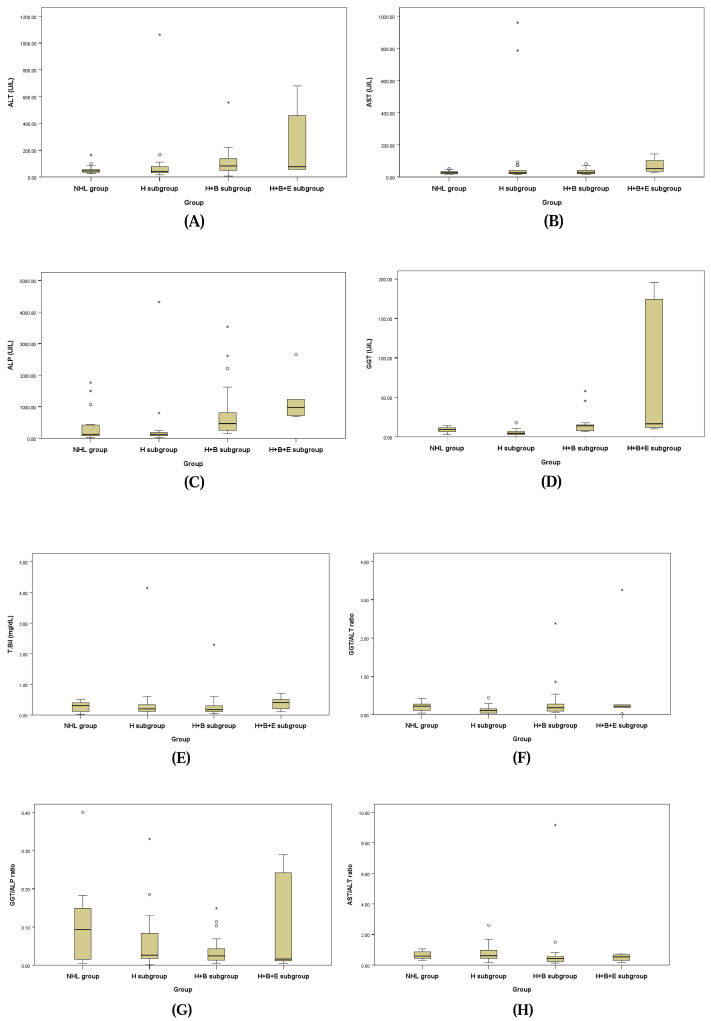
Box plots of ALT (**A**), AST (**B**), ALP (**C**), GGT (**D**), T. Bil (**E**), GGT/ALT ratio (**F**), GGT/ALP ratio (**G**), and AST/ALT ratio (**H**).

**Figure 2 animals-15-00256-f002:**
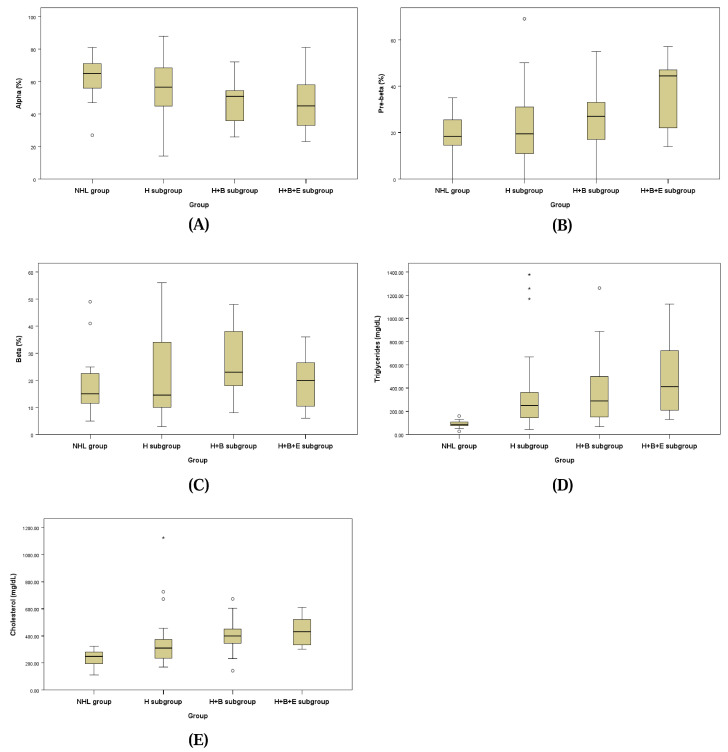
Box plots of alpha fraction (**A**), pre-beta fraction (**B**), beta-fraction (**C**), serum triglycerides (**D**) and cholesterol (**E**).

**Figure 3 animals-15-00256-f003:**
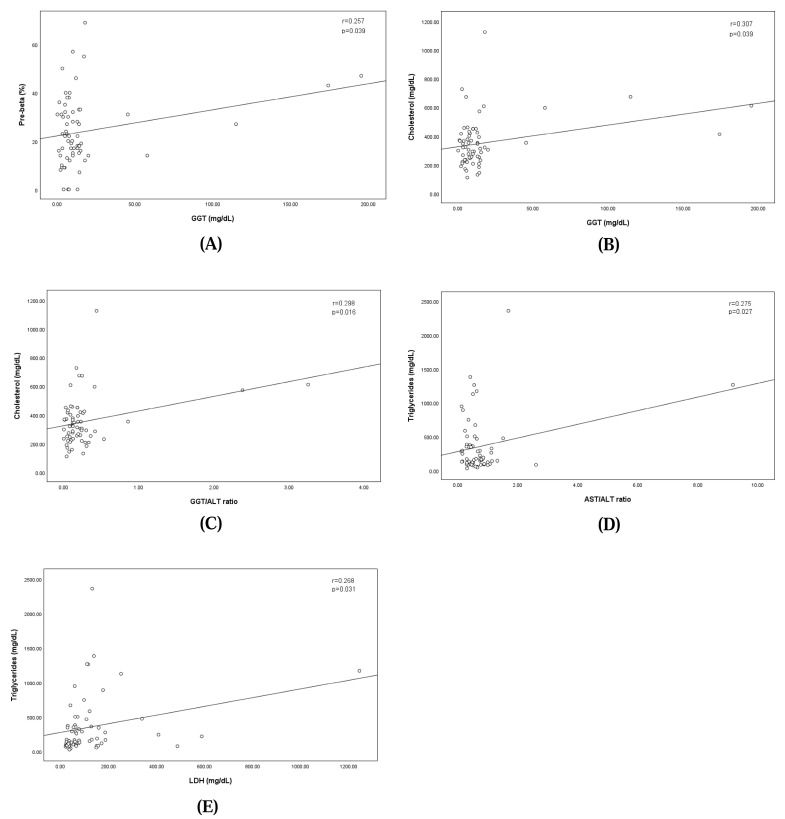
Linear relationships between pre-beta fraction and the concentration of serum GGT (**A**), between concentrations of serum GGT and Cholesterol (**B**), between concentrations of serum GGT/ALT ratio and Cholesterol (**C**), between concentrations of serum AST/ALT ratio and Triglycerides (**D**), and between concentrations of serum LDH and Triglycerides (**E**).

**Table 1 animals-15-00256-t001:** Signalment of the study population (*n* = 65).

Variable	NHL Group (*n* = 16)		HL Group (*n* = 49)		*p*
	*n*	%	*n*	%	
Sex					
IF	5	31.3%	12	24.5%	0.664
SF	3	18.8%	15	30.6%	
IM	3	18.8%	5	10.2%	
CM	5	31.3%	17	34.7%	
Age					
~ 4 Years	3	18.8%	4	8.2%	0.491
5~10 Years	9	56.3%	32	65.3%	
11 Years ~	4	25.0%	13	26.5%	
Breed					
Miniature Schnauzer	0	0.0%	18	36.7%	0.026 *
Yorkshire Terrier	4	25.0%	6	12.2%	
Maltese	5	31.3%	4	8.2%	
Shih Tzu	3	18.8%	6	12.2%	
Chihuahua	2	12.5%	2	4.1%	
Mixed	0	0.0%	4	8.2%	
Others ^a^	2	12.5%	4	10.0%	
Body condition score (BCS)					
(five-point system)					
1/5	0	0.0%	1	2.0%	0.146
2/5	1	6.2%	0	0.0%	
3/5	2	12.5%	13	26.5%	
4/5	4	25.0%	19	38.8%	
5/5	9	56.2%	16	32.7%	

^a^ Others: Poodle, Cocker Spaniel, Golden Retriever, Pomeranian. Differences for all analysis were considered to be significant at * *p* < 0.05 by the Chi-square (χ2) test. The data are expressed as number (*n*) and percentage (%). NHL group: Non-hyperlipidemic group; HL group: Hyperlipidemic group; IF: Intact female; SF: Spayed female; IM: Intact male; CM: Castrated male.

**Table 2 animals-15-00256-t002:** Clinical signs of the study subjects (*n* = 65).

Clinical Sign	NHL Group (*n* = 16)		HL Group (*n* = 49)		*p*
	*n*	%	*n*	%	
Anorexia	0	0.0%	4	8.2%	0.238
Vomiting	2	12.5%	9	18.4%	0.587
Diarrhea	0	0.0%	1	2.0%	0.565
Abdominal Distension	0	0.0%	13	26.5%	0.021 *
Abdominal Pain	2	12.5%	11	22.4%	0.388
Neurologic Signs	3	18.8%	5	10.2%	0.366
Ophthalmologic Signs	2	12.5%	5	10.2%	0.797
Cutaneous Signs	1	6.3%	4	8.2%	0.803

The data are expressed as number (*n*) and percentages (%). Differences for all analysis were considered to be significant at * *p* < 0.05 by the Chi-square (χ2) test. NHL group: Non-hyperlipidemic group; HL group: Hyperlipidemic group.

**Table 3 animals-15-00256-t003:** Clinical diagnosis of the study subjects (*n* = 65).

Clinical Diagnosis	NHL Group (*n* = 16)		HL Group (*n* = 49)		*p*
	*n*	%	*n*	%	
Obesity	13	81.2%	35	71.4%	0.602
Hypothyroidism	2	12.5%	4	8.2%	0.271
HAC	1	6.2%	4	8.2%	0.062
PLN	0	0.0%	1	2.0%	0.332
DM	1	6.2%	5	10.2%	0.225
Pancreatitis	1	6.2%	6	12.2%	0.451
Others ^a^	12	75.0%	42	85.7%	0.985

^a^ Others: urinary bladder calculi, mitral valve insufficiency, renal calculi, pododermatitis, tracheal collapse, chronic kidney disease, hydrocephalus, cataract, atopic dermatitis, allergic dermatitis, soft palate elongation, periodontitis, Evans syndrome, tricuspid valve insufficiency, lipoma, lymphoma, pulmonary hypertension, intervertebral disk disease, urinary incontinence, keratoconjunctivitis sicca, gastric foreign body, progressive retinal atrophy, protein-losing enteropathy, canine distemper virus infection, pyoderma, xylitol intoxication, nasal cell carcinoma, cataplexy, perianal gland epithelioma. The data are expressed as numbers (*n*) and percentages (%). Differences for all analysis were considered to be significant at *p* < 0.05 by the Chi-square (χ2) test. NHL group: Non-hyperlipidemic group; HL group: Hyperlipidemic group; HAC: Hyperadrenocorticism; PLN: Protein-losing nephropathy; DM: Diabetes mellitus.

**Table 4 animals-15-00256-t004:** Complete Blood Count (CBC) of the study subjects (*n* = 65).

Variable	NHL Group (*n* = 16)		HL Group (*n* = 49)		Reference Range	*p*
	M	SD	M	SD		
WBC (×10^3^/μL)	14.65	7.83	14.12	11.11	6 ~ 17	0.862
RBC (×10^3^/μL)	7.19	1.00	7.06	1.02	5.5 ~ 8.5	0.679
HGB (gm/dL)	17.13	2.58	17.09	2.83	12 ~ 18	0.959
PCV (%)	47.79	5.70	47.44	6.62	37 ~ 55	0.851
MCV (fL)	66.75	4.02	67.21	4.76	60 ~ 74	0.731
MCHC (%)	23.79	2.11	24.30	3.67	31 ~ 36	0.599
PLT (×10^3^/μL)	35.32	3.38	35.68	5.52	200 ~ 500	0.806

Differences for all analysis were considered to be significant at *p* < 0.05. The data are expressed as mean (M) and standard deviation (SD) values. NHL group: Non-hyperlipidemic group; HL group: Hyperlipidemic group; PLT: Platelet.

**Table 5 animals-15-00256-t005:** Serum chemistry of the study subjects (*n* = 65).

Variable	NHL Group (*n* = 16)		HL Group (*n* = 49)		Reference Range	*p*
	M	SD	M	SD		
BUN (mg/dL)	18.32	12.52	17.60	7.85	8 ~ 26	0.785
CRE (mg/dL)	0.71	0.62	0.76	0.24	0.5 ~ 1.3	0.634
ALT (U/L)	54.50	35.36	244.80	880.08	19 ~ 70	0.393
AST (U/L)	30.00	9.67	73.20	169.62	15 ~ 43	0.315
ALP (U/L)	396.06	553.55	704.90	1013.21	15 ~ 127	0.250
GGT (mg/dL)	9.21	3.61	19.62	38.94	0 ~ 6	0.292
LDH (U/L)	87.56	114.76	133.73	192.82	0 ~ 130	0.369
CK (U/L)	143.63	153.99	225.71	344.41	46 ~ 320	0.361
T. Chol (mg/dL)	231.69	62.37	389.67	172.57	135 ~ 345	0.001 *
TG (mg/dL)	89.56	31.60	435.78	449.64	19 ~ 133	0.003 *
T. Bil (mg/dL)	0.25	0.15	0.37	0.65	0 ~ 0.344	0.511
Lipase (U/L)	71.67	45.80	184.61	261.38	0 ~ 500	0.303
Glucose (mg/dL)	129.75	68.25	153.55	115.91	70 ~ 118	0.441
TP (g/dL)	6.39	0.91	6.77	0.73	5.4 ~ 7.4	0.102
ALB (g/dL)	3.39	0.48	3.59	0.63	2.9 ~ 4.2	0.253
Ca (mg/dL)	10.89	1.26	10.98	1.27	8.8 ~ 11	0.799
P (mg/dL)	3.38	0.93	3.57	1.25	3 ~ 6.2	0.589
NH3 (μmol/L)	63.91	24.12	66.77	33.54	16 ~ 75	0.804
K (mmol/L)	4.15	0.55	4.17	0.68	3.5 ~ 5.0	0.945
Na (mmol/L)	147.00	2.68	145.41	6.22	141 ~ 152	0.326
Cl (mmol/L)	110.44	6.44	108.76	8.15	102 ~ 117	0.455

Differences for all analysis were considered to be significant at * *p* < 0.05. The data are expressed as mean (M) and standard deviation (SD) values. NHL group: Non-hyperlipidemic group; HL group: Hyperlipidemic group; TG: Triglycerides; T. Bil: Total bilirubin; TP: Total protein; ALB: Albumin; NH3: Ammonia; Na: Natrium; K: Kalium; Cl: Chloride.

**Table 6 animals-15-00256-t006:** Hepatobilliary profiles of the study subjects (*n* = 65).

Group	*n*	ALT (U/L)	AST (U/L)	ALP (U/L)	GGT (U/L)
NHL group ^a^	16	54.50 ± 35.36	30.00 ± 9.67	396.06 ± 553.55	9.21 ± 3.61
HL group		244.80 ± 880.08	73.20 ± 169.62	704.90 ± 1013.21	19.62 ± 3.61
H subgroup ^b^	24	348.00 ± 1250.84	103.33 ± 239.23	320.80 ± 864.85	5.13 ± 3.86
H + B subgroup ^c^	19	118.16 ± 122.22	36.47 ± 19.05	1031.05 ± 1111.59	21.75 ± 26.08
H + B + E subgroup ^d^	6	233.00 ± 269.12	69.00 ± 46.13	1208.50 ± 741.40	70.80 ± 88.70
*p*-Value		0.647	0.370	0.019 *	0.000 *
**Group**	** *n* **	**T. Bil (U/L)**	**GGT/ALT**	**GGT/ALP**	**AST/ALT**
NHL group ^a^	16	0.26 ± 0.15	0.212 ± 0.113	0.105 ± 0.105	0.655 ± 0.258
HL group		0.37 ± 0.65	0.271 ± 0.559	0.057 ± 0.074	0.786 ± 1.305
H subgroup ^b^	24	0.39 ± 0.81	0.115 ± 0.105	0.061 ± 0.075	0.770 ± 0.535
H + B subgroup ^c^	19	0.34 ± 0.51	0.334 ± 0.533	0.039 ± 0.041	0.899 ± 2.030
H + B + E subgroup ^d^	6	0.38 ± 0.21	0.696 ± 1.258	0.096 ± 0.132	0.492 ± 0.230
*p*-Value		0.916	0.054	0.105	0.867

The data are expressed as mean ± SD values. *: Significant difference at *p* < 0.05 levels among groups by one-way analysis of variance (ANOVA). Significant difference of magnitude among groups (*p* < 0.05) by Mann–Whitney test. ALT: d > a, c > a, d > b, c > b; ALP: d > a, c > a, d > b, c > b; GGT: d > a, c > a, b > a, c > b, d > b, a > b; GGT/ALT: c > b, a > b; AST/ALT: c > a, c > b; NHL group: Non-hyperlipidemic group; HL group: Hyperlipidemic group; H subgroup: Hyperlipidemia without biliary tract disease; H + B subgroup: Hyperlipidemia with biliary tract disease and without endocrine disease; H + B + E subgroup: Hyperlipidemia with both biliary tract disease and endocrine disease.

**Table 7 animals-15-00256-t007:** Lipid fraction profiles of the study subjects (*n* = 65).

Group	*n*	Lipoprotein Fraction (%)			Triglycerides (mg/dL)	Cholesterol (mg/dL)
		Alpha	Pre-Beta	Beta		
NHL group ^a^	16	62.50 ± 14.170	19.13 ± 8.563	18.50 ± 11.944	89.56 ± 31.595	231.69 ± 62.365
HL group						
H subgroup ^b^	24	56.29 ± 17.568	22.63 ± 15.871	21.17 ± 15.319	449.71 ± 551.762	364.92 ± 210.875
H + B subgroup ^c^	19	48.16 ± 14.163	25.21 ± 13.265	26.84 ± 12.668	458.53 ± 369.402	408.32 ± 133.399
H + B + E subgroup ^d^	6	47.50 ± 20.285	38.17 ± 16.510	14.50 ± 6.442	312.33 ± 157.660	429.67 ± 105.341
*p*-Value		0.048 *	0.040 *	0.143	0.026 *	0.006 *

*: Significant difference at *p* < 0.05 levels among groups by one-way analysis of variance (ANOVA). The data are expressed as mean ± SD values. Significant difference of magnitude among groups (*p* < 0.05) by post hoc comparisons. Alpha: a > c; Pre-beta: d > a, d > b, d > c; Triglyceride: b > a, c > a; Cholesterol: b > a, c > a, d > a. NHL group: Non-hyperlipidemic group; HL group: Hyperlipidemic group; H subgroup: Hyperlipidemia without biliary tract disease; H + B subgroup: Hyperlipidemia with biliary tract disease and without endocrine disease; H + B + E subgroup: Hyperlipidemia with both biliary tract disease and endocrine disease.

**Table 8 animals-15-00256-t008:** Lipid profiles and underlying diseases.

Group	*n*	Lipoprotein Fraction (%)			Triglycerides (mg/dL)	Cholesterol (mg/dL)
		Alpha	Pre-Beta	Beta		
NHL group ^a^	16	62.50 ± 14.170	19.1 ± 8.563	18.50 ± 11.944	89.56 ± 31.595	231.69 ± 62.365
Obesity ^b^	32	61.63 ± 15.545	18.44 ± 11.873	20.03 ± 13.020	346.41 ± 477.127	307.16 ± 135.630
Hypothyroidism ^c^	4	52.75 ± 20.255	33.25 ± 16.008	14.25 ± 6.946	234.25 ± 211.571	291.25 ± 145.151
HAC ^d^	4	39.50 ± 7.047	39.50 ± 6.608	21.00 ± 8.524	406.00 ± 485.953	414.25 ± 137.967
PLN ^e^	1	60.00	16.00	23.00	80.00	370
DM ^f^	5	52.20 ± 19.867	26.40 ± 13.297	22.00 ± 17.944	435.00 ± 336.621	415.20 ± 120.759
Pancreatitis ^g^	6	51.00 ± 4.967	19.25 ± 9.811	30.00 ± 10.231	507.50 ± 553.071	375.25 ± 133.859
*p*-Value		0.102	0.008 *	0.706	0.311	0.032 *

*: Significant difference at *p* < 0.05 levels among groups by one-way analysis of variance (ANOVA). The data are expressed as mean ± SD values. Significant difference of magnitude among groups (*p* < 0.05) by post hoc comparisons. Pre-beta: c > b, d > b, c > a, d > g, d > a; Cholesterol: b > a, d > a, f > a, g > a; NHL group: Non-hyperlipidemic group; HAC: Hyperadrenocorticism; PLN: Protein-losing nephropathy; DM: Diabetes mellitus.

**Table 9 animals-15-00256-t009:** Lipid profiles of 18 schnauzers.

Dog No.	Group	Diagnosis	Lipoprotein Fraction (%)			Triglycerides (mg/dL)	Cholesterol (mg/dL)
			Alpha	Pre-Beta	Beta		
1	Subgroup 2	Perinal gland Epithelioma	26	32	42	500	450
2	Subgroup 3	DM	46	46	8	500	450
3	Subgroup 2	MVI, epilepsy	54	18	28	380	345
4	Subgroup 2	Renal calculi	52	40	8	164	418
5	Subgroup 1	Renal calculi	87	0	13	186	274
6	Subgroup 1	Evans syndrome	59	32	9	241	234
7	Subgroup 1	Lipoma	39	8	53	1258	415
8	Subgroup 1	MVI Renal calculi	26	31	43	2354	726
9	Subgroup 3	HAC	34	33	33	1125	312
10	Subgroup 1	Lymphoma	45	17	38	370	292
11	Subgroup 1	cPRA	66	30	4	42	364
12	Subgroup 2	MVI Renal calculi	38	38	23	348	460
13	Subgroup 2	Corneal ulcer KCS	52	0	48	887	430
14	Subgroup 1	Gastric FB	58	22	20	342	250
15	Subgroup 1	cPRA	44	50	6	1168	320
16	Subgroup 1	Xylitol intoxication	69	17	14	217	367
17	Subgroup 2	Lipoma Pancreatitis	44	19	38	583	286
18	Subgroup 1	Nasal cell carcinoma	47	17	36	258	266
Average			49.2 ± 15.22	25.0 ± 14.37	25.7 ± 16.05	606.8 ± 69.12	369.9 ± 15.85
NHL group ^b^			62.5 ± 3.54	19.1 ± 2.14	18.5 ± 2.99	89.5 ± 7.89	231.6 ± 15.59
*p*-Value			0.013 *	0.164	0.148	0.001 *	0.000 *

*: Significant difference at *p* < 0.05 levels among groups by independent t-test. The data are expressed as mean ± SD values. Significant difference of magnitude between Schnauzer group and NHL group (*p* < 0.05). Alpha: Schnauzer group > NHL group. Triglyceride: Schnauzer group > NHL group. Cholesterol: Schnauzer group > NHL group. NHL group: Non-hyperlipidemic group; DM: Diabetes mellitus; MVI: Mitral valve insufficiency; HAC: Hyperadrenocorticism; cPRA: Central progressive retinal atrophy; KCS: Keratoconjunctivitis sicca; FB: Foreign body.

## Data Availability

The datasets used and analyzed during the current study are available from the corresponding author on reasonable request.

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
