# Peer review of "Lipid Metabolism Alterations in Hyperlipidemic Dogs with Biliary Tract or Endocrine Diseases"

_animals, 2025, doi:10.3390/ani15020256_

Round 1

Reviewer 1 Report

Comments and Suggestions for Authors

the work is interesting and innovative in the idea, but it needs to be improved in many parts

Author Response

[Simple summary and Abstract]

Comment 1:

Line 21,38: defining “screening tool” the LPE may be not correct, since actually it represent a tecnique which (for the cost and methodology) is not currently easily applicable in daily clinical practice. Maybe refer to an “useful” tool may be more appropriate.

Response 1:

According to the comment reviewer provided, we revised the expression 'screening tool' in the manuscript to 'useful tool'.

[Introduction]

Comment 2:

Line 63: and/or is better. Check bibliographical sources.

Response 2:

To clarify the meaning of the part, we checked the references again and modified the expression of the part to and/or according to the reviewer's comment.

Comment 3:

line 65: “Cholestasis, characterized by bile duct obstruction…”. The concept of “cholestasis” is wider, while here you only refer to extrahepatic cholestasis and obstruction. I recommend to use a more appropriate definition of cholestasis.

Response 3:

Since it is cholestasis characterized by bile duct obstruction, it is necessary to clarify it as extrahepatic cholestasis, as commented by the reviewer, and the expression has been corrected accordingly.

[Materials and methods]

Comment 4:

Lines 78-79: When were the samples processed? 12 years of storage is a lot, especially considering the storage at not -80°C. This may have a^ected the results. Authors should add literature on analyte stability and clarify these aspects.

Response 4:

For case selection, we reviewed clinical records of 65 animals that underwent LPE over a 12-year period. The LPE test for each animal was not conducted all at once after the samples were stored for up to 12 years but was conducted individually by requesting a commercial laboratory (Neodin Veterinary Science Institute) during the initial diagnosis process after each animal was admitted to the hospital. Blood samples (serum), as described in '2.6. Methodology and Evaluation of Lipoprotein Electrophoresis (LPE)', were stored at -20°C and sent to the testing agency within one day.

Considering these procedures, we believe the impact of sample processing on the LPE results is minimal.

Comment 5:

Line 90: you also excluded dogs with some ongoing treatments?

Response 5:

Before treating the dog, we performed LPE along with other clinical tests to determine the animal's health status and diagnosis. Treatment was carried out after clinical diagnosis, and therefore, we believe that the ongoing treatment has no effect on the LPE test results.

Comment 6:

Line 121-123: you did not consider total bilirubin as parameter of cholestasis? Can you please more precisely specify which ultrasonographic alterations you considered to consider a dog as presenting a “biliary tract disease”?

Response 6:

In evaluating biliary tract disease, we first identified increases in GGT and ALP levels during serum chemistry testing, alongside specific details from the history and physical examination. We then reviewed ultrasound findings of the liver and bile ducts and the reviewer's comments. Total bilirubin results, indicating extrahepatic hyperbilirubinemia, were referenced. Key ultrasound findings included bile duct dilatation, thickening of the bile duct wall, and mucocele. While preparing this manuscript, there were parts of the record that were missing, and these parts were added through this revision.

Comment 7:

Line 124-125: It is not clear how tests for endocrinopathies have been used. Please specify.

Response 7:

Based on the reviewer's comments, the tests and criteria for diagnosing endocrinopathies were revised to be clearer.

[Results]

Comment 8:

Line 182-…..: This part on clinical presentation is incorrect and confuses the study message. It is not useful to put the prevalence of symptoms, but the clinical diagnosis of cases included. It is necessary to add a table with the diagnoses of cases included in the two groups. I think this provides important information for the interpretation of the lipid pattern.

Response 8:

According to the reviewer's comments, we organized the clinical diagnoses for animals in the NHL and HL groups by adding Table 3. However, as the diagnoses confirmed through the treatment of each animal are highly diverse and can be influenced by factors such as age, diet, gender, and living environment, the frequencies and proportions of underlying conditions used in this study were summarized and analyzed using the chi-square test. Other diagnoses were collectively categorized as "Other," and only the total frequency, the proportion for each group, and the diagnosis names were recorded in the footnote below Table 3. Additionally, reflecting the added content in Table 3, the section "3.2. Clinical Presentations" was updated. With the inclusion of the new Table 3, the numbering of Tables 3 to 7 in the original manuscript was revised to Tables 4 to 8.

Comment 9:

Figure1: can you please use the terms you use in the M&M for the subgroups? Using “Subgroup 1-2-3” may cause confusion. Can you please put the letters a, b, c, d as over/underline characters?

Response 9:

Based on the reviewer's comments, the group names in Figure 1 have been revised from the original "Control Group," "Subgroup 1," "Subgroup 2," and "Subgroup 3" to align with the terminology used in the text: NHL Group, H Subgroup, H+B Subgroup, and H+B+E Subgroup.

Comment 10:

Lines 244-245: the statistical test applied is not clear . Please specify.

Response 10:

The contents of the relevant footnotes in Table 5­–7. were updated to reflect the reviewer’s comments, and statistically significant differences between each group were organized for clarity and ease of identification.

Comment 11:

259: “were associated” may not be the correct term considering the statistical test applied.

Response 11:

According to the comment reviewer provided, we revised the expression 'were associated with' in the manuscript to 'were correlate with'."

Comment 12:

261-262: same as above for the term “positive relationship”

Response 12:

According to the comment reviewer provided, we revised the expression 'positive relation' in the manuscript to 'positive correlation'.

Comment 13:

289: please specify in M&M how did you diagnose the pancreatitis, if acute or chronic, and which diagnostic imagining and laboratory criteria you used. Same for PLN.

Response 13:

Based on the reviewer's comments, the tests and criteria for diagnosing pancreatitis and protein-losing nephropathy were revised to be clearer.

Comment 14:

Table 6. Can you please put the letters a-g as over/underline characters?

Response 14:

While preparing the manuscript, we attempted to overline characters a–g after each disease name; however, the superscript formatting was inadvertently omitted. This issue has been corrected by the reviewer’s comments.

Comment 15:

Lines 293-297: The statistical test applied is not clear, but above, all the statistical results as they are put in the table are not clear. Another presentation which is easier to read should be used.

Response 15:

The contents of the relevant footnotes in Table 5­–7. were updated to reflect the reviewer’s comments, and statistically significant differences between each group were organized for clarity and ease of identification.

Comment 16:

Table 7: as above

Response 16:

In the case of Table 7, since it involves comparing significant differences between the Schnauzer group and the NHL group, separate overline characters were deemed unnecessary. Therefore, the relevant characters were removed, and the remaining footnote was revised accordingly.

[Discussion]

Comment 17:

333-334: I think that LPE is useful for the pathogenic investigation of hyperlipemia, which may have led to a shift in clinical management and therapy, not in diagnosis.

Response 17:

We appreciate the valuable feedback provided by the reviewer. We also agree with the reviewer’s comment that LPE is a useful test for investigating the pathogenesis of hyperlipidemia. However, from the perspective of a veterinary profession, it is essential to provide therapy and manage the clinical progression of animals with hyperlipemia. In this regard, we believe that the LPE test can offer valuable and precise information in the diagnostic process to assess the patient’s health status scientifically and accurately hyperlipemia.

Comment 18:

363: please see the comment above about the term “initial screening tool”. LPE may be a more useful tool in the diagnostic process and long-term monitoring of these patients rather than a screening tool.

Response 18:

According to the comment reviewer provided, we revised the expression 'screening tool' in the manuscript to 'useful tool'.

Comment 19:

371: You should also add the fact that your population did not included a healthy control group, and that the diagnosis of biliary tract disease was not further assessed or eventually confirmed by an hystological hepatic assessment. Not having the information regarding the diet and therapies, as well as the lack of their standardization also represent a limit of the present study, not only a future perspective.

Response 19:

At the Veterinary Medical Teaching Hospital (VMTH) of Konkuk University, where this study was conducted, lipid profile evaluation (LPE) tests are performed by an external commercial laboratory upon the veterinarian’s recommendation when deemed necessary. Only animals that underwent LPE were included for this study based on the inclusion criteria. Consequently, very few cases of LPE were being performed on healthy animals, making it challenging to establish a separate healthy control group. However, we have tried to rule out any abnormalities with a complete blood counts, serum chemistry profiles of each dogs to include normal dogs as a healthy group.

Additionally, for the 65 dogs included in this study, we reviewed data such as diet, living environment, chief complaints, physical examination results, TPR (temperature, pulse, respiration), and clinical pathology findings. However, it didn't prove easy to standardize and incorporate all of these factors due to variations in each dog's physiological and environmental conditions. As a result, rather than attempting to standardize these aspects, we designed the study to classify the dogs into the NHL group and HL group based on the presence of hyperlipidemia and conducted comparative analyses accordingly.

Thank you for your comment, we will take it into consideration in future research.

[Conclusions]

Comment 20:

382-388: These conclusions are vague and overall not strictly pertinent to the results and aim of this study. The absence of specific diagnoses in the two groups does not allow this conclusion.

Response 20:

In response to the reviewer's comments, the results of this study were summarized and organized, and a conclusion was added.

Regarding the clinical presentation, as per the reviewer's suggestions, we organized the clinical diagnoses for animals in the NHL and HL groups by adding Table 3, which was reflected in the updated content of section "3.2. Clinical Presentations." However, due to the diverse nature of clinical diagnoses observed in all animals included in the study, there were limitations in summarizing all diagnoses comprehensively. While the frequency and proportion for each group were summarized only for the underlying conditions used in this study, the chi-square test analysis did not yield statistically significant results. Consequently, these findings were not included separately in the Conclusion.

[References]

Comment 21:

The references cite works from 50 years ago which are absolutely inappropriate. Except for justified exceptions, the bibliography of an innovative work should not exceed 15-20 years

Response 21:

When preparing the manuscript for this paper and searching references on lipids, lipid electrophoresis, and secondary hyperlipoproteinemia, we extensively referred to and quoted portions from three seminal papers: 1 published in 1966 by Robert, I.L.; Robert, S.L.; Donald, S.F. and 2 published in 1975 by Rogers, W.A.; Donovan, E.F.; and Kociba, G. J. To the best of my knowledge, subsequent studies in this field heavily cite these three pioneering works. Rather than referencing later publications, we decided it would be more appropriate to cite these foundational papers directly, despite these works being from 50 years ago, as they are highly relevant to the specific focus of this study.

Reviewer 2 Report

Comments and Suggestions for Authors

Thank you for submitting your manuscript.

I think it is well written.

Minor revision

Please add about obesity information (BCS) each of the NHL group and HL group.

Author Response

Comment 1:

Please add about obesity information (BCS) each of the NHL group and HL group.

Response 1:

Based on the reviewer's comments, the frequency and ratio of BCS scores for each animal in the NHL group and HL group, along with the analysis results from the chi-square test, are additionally presented in Table 1. Furthermore, Section 3.1 has been updated to include the additions to the Signalment.

Reviewer 3 Report

Comments and Suggestions for Authors

General comments: This paper is well written and the authors did a lot of experiment which deserve praise but I want to point out few points.

Line 14-15 ‘such as endocrine diseases, exhibited higher pre-beta fractions and cholesterol concentrations compared to non-hyperlipidemic dogs’

As you know, pre-beta fraction stands for VLDL.

Line 20-21 ‘These findings highlight the clinical relevance of LPE as a screening tool for identifying lipid metabolism abnormalities and monitoring disease-related changes.’

You can calculate VLDL with ‘Friedewald equation’ as below.

dividing the triglyceride value by 5 equals VLDL

Line 103 ‘Cobas C III instrument (Roche Diagnostics, South San Francisco, CA, USA)’

You can measure HDL with Cobas instrument as below.

https://www.biolab-srl.com/products/cobas-hdl-cholesterol/

Combining these factors together, if you utilize blood chemistry equipment to measure HDL and calculate as ‘LDL = Total cholesterol - HDL – VLDL’,

We don’t have to utilize lipoprotein electrophoresis which cost additional fee to clients.

Therefore, there should be more reason why practitioner have to pay attention to LPE.

Same as line 22-23 ‘Veterinary practitioners should consider altered lipid metabolism as a potential indicator of biliary tract or endocrine diseases’, maybe it can be more detailed such as providing cut-off value of GGT/ALT ratio or GGT/ALP ratio in endocrine disease or biliary tract disease.

You can check this reference as below.

Yang, J. G., He, X. F., Huang, B., Zhang, H. A., & He, Y. K. (2018). Rule of changes in serum GGT levels and GGT/ALT and AST/ALT ratios in primary hepatic carcinoma patients with different AFP levels. Cancer biomarkers21(4), 743-746.

As you know that alpha fraction stands for HDL,

Line 280-281 ‘The alpha fraction of lipoproteins was lower in groups with underlying diseases compared to the NHL group; however, this difference was not statistically significant.’

Have you checked TC/HDL ratio as in human reference as below? I think there could be interesting results as in humans.

Lemieux, I., Lamarche, B., Couillard, C., Pascot, A., Cantin, B., Bergeron, J., ... & Després, J. P. (2001). Total cholesterol/HDL cholesterol ratio vs LDL cholesterol/HDL cholesterol ratio as indices of ischemic heart disease risk in men: the Quebec Cardiovascular Study. Archives of internal medicine161(22), 2685-2692.

Jeppesen, Facchini, & Reaven. (1998). Individuals with high total cholesterol/HDL cholesterol ratios are insulin resistant. Journal of internal medicine243(4), 293-298.

I cannot find the discussion of lipid profile in Schnauzers.

Line 311-312 ‘The mean alpha fraction in the 18 Schnauzers (49.22 ± 15.22%) was significantly lower than that in the NHL group (62.50 ± 3.54%; p = 0.013). ‘

This could mean schnauzers have lower HDL than other breed and HDL is known as good cholesterol which removes lipid from blood vessel so this could be interesting explanation for genetic hyperlipidemia although there are multiple factors such as SPINK1 mutation.

How about LPL with schnauzers in molecular level such as below reference?

Schickel, R. (2005). Identification of the nucleotide sequence of the lipoprotein lipase gene as well as its role in the development of hyperlipidemia and pancreatitis in the Miniature Schnauzer (Doctoral dissertation, lmu).

Or maybe you can check this reference also,

Furrow, E., Jaeger, J. Q., Parker, V. J., Hinchcliff, K. W., Johnson, S. E., Murdoch, S. J., ... & Brunzell, J. D. (2016). Proteinuria and lipoprotein lipase activity in Miniature Schnauzer dogs with and without hypertriglyceridemia. The Veterinary Journal212, 83-89.

It would be more significant and attractive research if you discuss schnauzers’ part more deeply.

Author Response

Comment 1:

Line 14-15 ‘such as endocrine diseases, exhibited higher pre-beta fractions and cholesterol concentrations compared to non-hyperlipidemic dogs’

As you know, pre-beta fraction stands for VLDL.

Line 20-21 ‘These findings highlight the clinical relevance of LPE as a screening tool for identifying lipid metabolism abnormalities and monitoring disease-related changes.’

You can calculate VLDL with ‘Friedewald equation’ as below.

dividing the triglyceride value by 5 equals VLDL

Line 103 ‘Cobas C III instrument (Roche Diagnostics, South San Francisco, CA, USA)’

You can measure HDL with Cobas instrument as below.

https://www.biolab-srl.com/products/cobas-hdl-cholesterol/

Combining these factors together, if you utilize blood chemistry equipment to measure HDL and calculate as ‘LDL = Total cholesterol - HDL – VLDL’,

We don’t have to utilize lipoprotein electrophoresis which cost additional fee to clients.

Therefore, there should be more reason why practitioner have to pay attention to LPE.

Response 1:

We appreciate the valuable comments provided by the reviewer.

When we referred the LPE analysis to an external commercial laboratory, the lipid profile results were provided as alpha, pre-beta, and beta fractions. While we considered converting these fractions into VLDL, LDL, and HDL, we were unable to obtain detailed information on the specific methodology at that time. Therefore, this manuscript was prepared based on the results provided by the outsourced LPE test.

According to the information kindly shared by the reviewer, we understand that VLDL can be calculated using the Friedewald equation. However, our hospital did not measure HDL levels as part of the serum chemistry analysis during the study period using the Cobas C III instrument. In addition, since the LPE test for each animal was conducted by sending the serum directly to an external commercial laboratory for diagnosis at the time of the visit, there are limitations to performing additional tests, even though the HDL parameter is currently configured on the Cobas C III instrument. As a result, HDL data for the 65 dogs included in this study are unavailable.

Additionally, we cannot use the provided LDL calculation formula without HDL values, which require HDL input.

Once again, we sincerely appreciate the information provided by the reviewer. We will incorporate and reference this information in future validation studies.

Comment 2:

Same as line 22-23 ‘Veterinary practitioners should consider altered lipid metabolism as a potential indicator of biliary tract or endocrine diseases’, maybe it can be more detailed such as providing cut-off value of GGT/ALT ratio or GGT/ALP ratio in endocrine disease or biliary tract disease.

You can check this reference as below.

Yang, J. G., He, X. F., Huang, B., Zhang, H. A., & He, Y. K. (2018). Rule of changes in serum GGT levels and GGT/ALT and AST/ALT ratios in primary hepatic carcinoma patients with different AFP levels. Cancer biomarkers21(4), 743-746.

Response 2:

We have reviewed the references provided by the reviewer and were impressed by the application of the GGT/ALT ratio or GGT/ALP ratio in diagnosing endocrine diseases and biliary tract diseases in human medicine for more precise diagnostics. While we are inspired to incorporate such approaches into veterinary internal medicine practice and research, to the best of our knowledge, the interpretation and application of these ratios in diagnosing diseases in animals remain underexplored and require further development in the veterinary field.

For this reason, it is currently challenging to include the GGT/ALT or GGT/ALP ratio in this study. However, we will make efforts to utilize these ratios in future validation studies and other related research.

Comment 3:

As you know that alpha fraction stands for HDL,

Line 280-281 ‘The alpha fraction of lipoproteins was lower in groups with underlying diseases compared to the NHL group; however, this difference was not statistically significant.’

Have you checked TC/HDL ratio as in human reference as below? I think there could be interesting results as in humans.

Lemieux, I., Lamarche, B., Couillard, C., Pascot, A., Cantin, B., Bergeron, J., ... & Després, J. P. (2001). Total cholesterol/HDL cholesterol ratio vs LDL cholesterol/HDL cholesterol ratio as indices of ischemic heart disease risk in men: the Quebec Cardiovascular Study. Archives of internal medicine161(22), 2685-2692.

Jeppesen, Facchini, & Reaven. (1998). Individuals with high total cholesterol/HDL cholesterol ratios are insulin resistant. Journal of internal medicine243(4), 293-298.

Response 3:

We have reviewed the references provided by the reviewer regarding the Total Cholesterol/HDL Cholesterol (TC/HDL) ratio versus the LDL Cholesterol/HDL Cholesterol (LDL/HDL) ratio and were impressed by their diagnostic application in human medicine, similar to the GGT/ALT or GGT/ALP ratios.

However, in the veterinary field, the interpretation and application of the TC/HDL ratio are still in the early stages of development. Additionally, as HDL levels were not measured in the animals included in this study, we could not calculate these ratios.

We sincerely appreciate the reviewer's sharing of this valuable information, and we will endeavor to incorporate the TC/HDL ratio into future validation studies and related research.

Comment 4:

I cannot find the discussion of lipid profile in Schnauzers.

Line 311-312 ‘The mean alpha fraction in the 18 Schnauzers (49.22 ± 15.22%) was significantly lower than that in the NHL group (62.50 ± 3.54%; p = 0.013). ‘

This could mean schnauzers have lower HDL than other breed and HDL is known as good cholesterol which removes lipid from blood vessel so this could be interesting explanation for genetic hyperlipidemia although there are multiple factors such as SPINK1 mutation.

How about LPL with schnauzers in molecular level such as below reference?

1) Schickel, R. (2005). Identification of the nucleotide sequence of the lipoprotein lipase gene as well as its role in the development of hyperlipidemia and pancreatitis in the Miniature Schnauzer (Doctoral dissertation, lmu).

 Or maybe you can check this reference also,

2)Furrow, E., Jaeger, J. Q., Parker, V. J., Hinchcliff, K. W., Johnson, S. E., Murdoch, S. J., ... & Brunzell, J. D. (2016). Proteinuria and lipoprotein lipase activity in Miniature Schnauzer dogs with and without hypertriglyceridemia. The Veterinary Journal212, 83-89.

 It would be more significant and attractive research if you discuss schnauzers’ part more deeply.

Response 4:

While preparing the manuscript, the discussion related to Schnauzer was omitted, so the discussion was added to the manuscript. In this process, the references provided by the reviewer and two related papers were referred to and cited, and then the discussion was added to these three papers. The relevant information of the cited three papers has been included in the 'References' section. We want to thank the reviewers who left critical comments and information, and we will strive to utilize genetic hyperlipidemia in Schnauzer in future validation studies and other studies.

Round 2

Reviewer 3 Report

Comments and Suggestions for Authors

The authors revised the manuscript well according to the reviewer's comments.

Still, there should be more analysis about cut-off value dealing with below comments.

It won't take time so long because the authors already have data and it can be achieved by using statistical software such as SPSS which is already used in the study.

'‘Veterinary practitioners should consider altered lipid metabolism as a potential indicator of biliary tract or endocrine diseases’,

-> maybe it can be more detailed such as providing cut-off value of GGT/ALT ratio or GGT/ALP ratio in endocrine disease or biliary tract disease.

Author Response

Comment 1: 
 The authors revised the manuscript well according to the reviewer's comments. Still, there should be more analysis about cut-off value dealing with below comments.

 It won't take time so long because the authors already have data and it can be achieved by using statistical software such as SPSS which is already used in the study.

'‘Veterinary practitioners should consider altered lipid metabolism as a potential indicator of biliary tract or endocrine diseases’,

-> maybe it can be more detailed such as providing cut-off value of GGT/ALT ratio or GGT/ALP ratio in endocrine disease or biliary tract disease.

Response 1: 
 We thank the reviewer for their valuable comments, which have provided an opportunity to further enhance the manuscript. In response to the suggestion regarding cut-off values for the GGT/ALT and GGT/ALP ratios, we conducted additional analyses using the SPSS software to examine these parameters in greater detail.

 We included hepatobiliary function markers (ALT, AST, ALP, GGT, and total bilirubin) and calculated the GGT/ALT, GGT/ALP, and AST/ALT ratios. The analysis revealed significant positive correlations between the GGT/ALT ratio and cholesterol levels, as well as between the AST/ALT ratio and triglyceride levels. Group differences for these parameters were also evaluated. However, despite these observations, the study did not demonstrate clear statistical significance between groups for the GGT/ALT ratio, GGT/ALP ratio, and AST/ALT ratio.

 The manuscript has been revised to reflect these findings. Specifically, we have included a new figure (Figure 1) and table (Table 6) to present the results, and the Results and Discussion sections have been updated accordingly. The revised sections in the manuscript have been highlighted using green marker for clarity.

 While the additional analysis did not yield statistically significant cut-off values, we believe that these ratios provide important insights into distinguishing between liver and gallbladder dysfunction in veterinary practice.